

# Repurposing FDA-approved phytomedicines, natural products, antivirals and cell protectives against SARS-CoV-2 (COVID-19) RNA-dependent RNA polymerase

Mahmoud Kandeel[1,2], Yukio Kitade[3,4] and Abdullah Almubarak[1]

[1] King Faisal University, Al-Ahsa, Al-Ahsa, Saudi Arabia
[2] Kafrelshikh University, Kafrelshikh, Kafrelshikh, Egypt
[3] Gifu University, Gifu, Japan
[4] Aichi Institute of Technology, Toyota, Aichi, Japan

Corresponding author
Mahmoud Kandeel,
mkandeel@kfu.edu.sa

## ABSTRACT

Following the recent emergence of SARS-CoV-2 or coronavirus disease 2019 (COVID-19), drug discovery and vaccine design to combat this fatal infection are critical. In this study, an essential enzyme in the SARS-CoV-2 replication machinery, RNA-dependent RNA polymerase (RDRP), is targeted in a virtual screening assay using a set of 1,664 FDA-approved drugs, including sets of botanical and synthetic derivatives. A set of 22 drugs showed a high docking score of >−7. Notably, approximately one-third of the top hits were either from natural products or biological molecules. The FDA-approved phytochemicals were sennosides, digoxin, asiaticoside, glycyrrhizin, neohesperidin, taxifolin, quercetin and aloin. These approved natural products and phytochemicals are used as general tonics, antioxidants, cell protectives, and immune stimulants (nadid, thymopentin, asiaticoside, glycyrrhizin) and in other miscellaneous systemic or topical applications. A comprehensive analysis was conducted on standard precision and extra precision docking, two-step molecular dynamics simulations, binding energy calculations and a post dynamics analysis. The results reveal that two drugs, docetaxel and neohesperidin, showed strong binding profiles with SARS CoV-2 RdRP. These results can be used as a primer for further drug discovery studies in the treatment of COVID-19. This initiative repurposes safe FDA-approved drugs against COVID-19 RdRP, providing a rapid channel for the discovery and application of new anti-CoV therapeutics.

## INTRODUCTION

Coronavirus disease 2019 (COVID-19) appeared in the Hubei prefecture of China in December 2019 and spread widely within a month, with cases recorded in more than two countries (WHO 2020, https://www.who.int/emergencies/diseases/novel-coronavirus-2019). The causative agent was determined to be a new coronavirus, SARS-CoV-2.

SARS-CoV-2 is a member of the *Betacoronavirus* genus (*Liu et al., 2020*), a group of RNA viruses characterized by the presence of a polyprotein and a relatively large genome (~30 kb) that codes for multiple structural and nonstructural proteins. The four structural proteins include spike (S), envelope (E), membrane (M) and nucleocapsid (N) (*Shi et al., 2015*). Approximately 14–16 nonstructural proteins are encoded at the 5′-terminal of the polyprotein. The proteins are essential for virus replication, mRNA synthesis, and processing of viral proteins (*Brian & Baric, 2005*). Coronaviruses have recently evolved from a causative agent of mild respiratory symptoms to severely pathogenic forms. Three major pathogenic and fatal CoVs have emerged in the last two decades: SARS CoV, MERS CoV, and lastly, SARS-CoV-2 (*Guarner, 2020*).

RNA-dependent RNA polymerase (RdRP) is a nonstructural protein that is crucial for SARS-CoV-2 replication (*Gaurav & Al-Nema, 2019*). RdRP is a significant drug target against CoVs, implemented for the discovery of new antivirals against MERS CoV (*Gordon et al., 2020*), dengue (*Shimizu et al., 2019*), Zika (*Ahmad et al., 2020*), influenza A (*Niazi et al., 2019*) and hepatitis C viruses (*Dash, Aydin & Stephens, 2019*).

In this study, SARS-CoV-2 RdRP was used to virtually screen a dataset of safe, non-toxic and FDA-approved pharmacological compounds. The drugs with a high score can be repurposed to treat SARS-CoV-2, taking advantage of its safety and direct applicability without further toxicity or preclinical testing. In addition, the phylogenetic relationships between SARS-CoV-2 RdRP and those of the viruses responsible for other recent coronavirus epidemics, SARS and MERS CoV, were investigated. Full molecular and dynamics profiles of drugs binding to RdRP were evaluated by combining docking and Molecular Dynamics (MD) simulations.

# MATERIALS AND METHODS

## Construction of FDA approved drugs dataset

A total of 1,664 FDA approved drugs were retrieved from commercial compounds databases at Selleck Inc., USA. All compounds were imported to Ligprep software, desalted and 3D optimized using OPLS2005 force field at physiological pH. The compounds dataset are provided in Supplemental Files 1.

## SARS-CoV-2 RdRP optimization and structure preparation

The protein preparation module in Maestro software package (Schrodinger LLC, NY, USA) was used to optimize the protein structure for docking. The protein (PDB ID 7BV2) was protonated, optimized at cellular pH conditions, and energy minimized using OPLS2005 force field. All ions, except the active site magnesium, and non-relevant crystallographic materials were removed. During preparation, two structural files were prepared comprising RdRP in the presence or absence of magnesium ions. The docking grid was generated by the selection of active site residues. The docking box was centered around the active site residues ASP255 and ASP256 with a size of 20 Å. For comparative purpose, the sequence similarity between SARS CoV, SARS CoV-2 and MERS CoV RdRPs were retrieved and aligned (Supplemental Methods File). The reliability

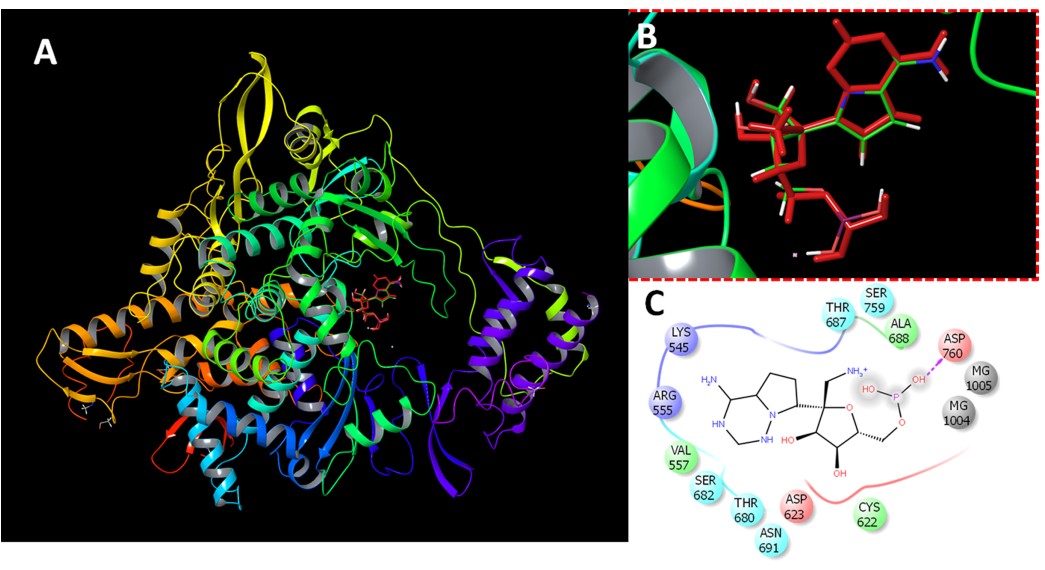

**Figure 1 Docking results.** Docking of remdesivir with COVID-19 CoV RdRP. (A) The docking site of remdesivir monophosphate into the active site of RdRP. (B) The docking site showing the superimposed conformations of the co-crystalized ligand and the docked pose. The docked pose is colored by red and the co-crystalized ligand is colored by the atom type. (C) The ligand interactions of remdesivir monophosphate with the active site of RdRP.                         

of docking procedures were assigned by redocking of the co-crystalized inhibitor (remdesivir). The obtained poses were comparable to the co-crystalized structure (Fig. 1).

## Virtual screening

The glide docking module of Schrodinger maestro package was used for virtual screening. Two-step docking protocol was carried out. At first, the docking was set to run under the standard precision (SP docking) (*Friesner et al., 2004*; *Halgren et al., 2004*). The results were ranked by the docking score. Remdesivir was used as standard RdRP inhibitor. It is a broad-spectrum antiviral with potent inhibition of MERS CoV RdRP (*Gordon et al., 2020*). Compounds with a docking score of −7.00 or higher were retrieved and reanalyzed by extra-precision docking protocol (XP-docking). Magnesium ions are important in coordinating the polymerase actions of RdRP. Therefore, the two docking interactions were run comprising the presence and absence of Mg ions in their binding sites.

## Molecular dynamics simulations

Groningen Machine for Chemical Simulations (GROMACS) 5.1.4. software was used in all molecular dynamics (MD) simulations (*Abraham et al., 2015*; *Van Der Spoel et al., 2005*). The ligand parameter, topology and restraint and the protein were handled by AMBERFF14SB force field and general AMBER force field (GAFF), respectively. The RdRP-ligand complexes were solvated in a cubic box with 1.0 nm from protein to box edge. The box was filled with a single point charge (SPC) water model. The solvated complexes were minimized for 5,000 steps. During water and ions coupling, the heavy atoms of protein and ligand were restrained. Two-step system equilibration was adopted comprising 50 ps NVT at 300 K followed by 1 ns NPT ensemble at 300 K. Production

stages were done over simulation times of 20 and 100 ns with NPT ensemble. Parrinello–Rahman algorithm maintain constant pressure at 1 bar and V-rescale thermostat algorithmfor temperature coupling at 300 K. Long-range electrostatics under periodic conditions with a direct space cut-off 12 Å were restrained by Particle Mesh Ewald (PME) method. Time step was set to 2 fs. Coordinates and output values were collected every 10 ps.

### Trajectory analysis

GROMACS MD simulation toolkits were used in the trajectory analysis. The g_rms and g_rmsf were used to calculate the root mean square deviation (RMSD) of the starting structure to the end of the simulation time and the per-residue root mean square fluctuation (RMSF) of the protein residues.

### Binding energy calculations

The molecular mechanics–generalized Born surface area (MM/GBSA) was calculated by the g_mmpbsa tool.

### Statistical analysis

Multiple correlation comparisons were performed between the obtained docking score and the properties and interactions of the compounds with SARS-CoV-2, including the molecular weight, the hydrogen bonding and the lipophilic interaction parameters. The analysis was carried out with the Pearson's correlation coefficient, implemented in GraphPad Prism version 7 (GraphPad Software, La Jolla, CA, USA).

## RESULTS

### Virtual screening and docking

The virtual screening data for 1,664 FDA-approved drugs against SARS-CoV-2 are provided in Table S1. The presented parameters include the docking scores, ligand efficiencies, and lipophilic and hydrogen bonding interactions. The assessment of docking protocol was confirmed by docking of remdesivir monophosphate (RMP). The redocked pose was compared with the co-crystalized ligand (Fig. 1). It is was evident that the presence of magnesium ions increased the docking scores of RMP in both of the standard precision and extra-precision docking. In the absence of magnesium ions, the SP and XP docking scores of RMP were −8.9 and −9.7, respectively. These scores was decreased by about 15% in the absence of magnesium ions. In contrast, the docked ligands from the FDA approved drugs dataset showed higher scores in the absence of magnesium ions. This might be attributed to the favorable interaction of magnesium ions with the phosphorylated substrates.

Table 1 shows the 26 compounds with the highest docking scores, which were −7 or higher. Within the top hits, there were five general tonics, cell protectives, and antioxidants; one antiviral; one anti-parasitic; one antibiotic; one immune stimulant; three anticancer drugs; and other miscellaneous systemically or topically acting drugs.

**Table 1 Natural compounds showing the highest docking score.**

| Name | Mol. weight | Docking score | Glide ligand efficiency | Glide lipo | Glide hbond | Glide evdw | Source | Clinical uses |
|---|---|---|---|---|---|---|---|---|
| Natural compounds showing the highest docking score (>−7) after docking of a set of FDA approved drugs against SARS-CoV-2 RNA-dependent RNA polymerase | | | | | | | | |
| Sennoside B | 862.7 | −8.1 | −0.1 | −0.6 | −0.2 | −58.8 | Glycoside from Senna plants | Constipation |
| Digoxin | 780.9 | −7.8 | −0.1 | −2.4 | −0.2 | −62.4 | Glycoside from digitalis | cardiovascular |
| Asiaticoside | 959.1 | −7.6 | −0.1 | −1.8 | 0.0 | −50.4 | triterpenoid from *Centella asiatica* | antioxidant |
| Glycyrrhizin (Glycyrrhizic Acid) | 822.9 | −7.6 | −0.1 | −0.8 | 0.0 | −55.6 | triterpene glycoside from licorice | Hepatoprotective, food sweetener |
| Neohesperidin dihydrochalcone (Nhdc) | 612.6 | −7.5 | −0.2 | −1.4 | −0.1 | −46.7 | flavanone glycoside in citrus fruits | food sweetener |
| Taxifolin (Dihydroquercetin) | 304.3 | −7.4 | −0.3 | −1.3 | −0.4 | −23.7 | Flavonoid present in many plants | anticancer |
| Quercetin (Sophoretin) | 302.2 | −7.1 | −0.3 | −1.1 | −0.4 | −25.1 | Flavonoid in citrus fruits | PI3K, PKC, Src, Sirtuin |
| Aloin (Barbaloin) | 418.4 | −7.0 | −0.2 | −1.5 | −0.3 | −28.3 | Glycoside from Aloe | Tyrosinase |

| Name | Mol. weight | Docking score | Glide ligand efficiency | Glide lipo | Glide hbond | Glide evdw | Clinical uses |
|---|---|---|---|---|---|---|---|
| Synthetic drugs showing the highest docking score (>−7) after docking of a set of FDA approved drugs against SARS-CoV-2 RNA-dependent RNA polymerase | | | | | | | |
| Nadid (NAD+) | 663.4 | −8.2 | −0.2 | −0.9 | −0.5 | −56.2 | Chronic fatigue/general tonic |
| Leucovorin | 601.6 | −8.1 | −0.2 | −0.3 | −1.5 | −32.2 | folate analog |
| Thymopentin | 679.8 | −7.9 | −0.2 | −1.6 | −1.5 | −50.3 | Immune stimulant & Inflammation related |
| Ritonavir | 720.9 | −7.7 | −0.2 | −2.9 | −0.3 | −67.9 | HIV Protease |
| Venetoclax | 868.4 | −7.5 | −0.1 | −2.5 | −0.6 | −64.0 | Bcl-2 inhibitor |
| Oxiglutatione | 612.6 | −7.4 | −0.2 | −0.8 | −0.8 | −48.2 | Cell protection |
| Iopamidol | 777.1 | −7.4 | −0.2 | −0.9 | −1.1 | −37.1 | Contrast agent |
| Acarbose | 645.6 | −7.3 | −0.2 | −1.2 | −0.3 | −37.2 | Type II diabetes |
| Chlorhexidine HCl | 578.4 | −7.3 | −0.2 | −1.9 | −0.4 | −39.7 | Topical antiseptic |
| Salvianolic acid B | 718.6 | −7.3 | −0.1 | −1.2 | 0.0 | −55.3 | Others |
| Troxerutin | 742.7 | −7.2 | −0.1 | −1.0 | −0.5 | −42.1 | Thrombin |
| Cefodizime Sodium | 628.6 | −7.2 | −0.2 | −1.0 | −0.9 | −44.0 | antibiotic |
| (−) Epicatechin | 290.3 | −7.1 | −0.3 | −1.0 | −0.6 | −22.0 | Others |
| Ioversol | 807.1 | −7.1 | −0.2 | −0.7 | −1.1 | −38.1 | Contrast dyes |
| Pyrantel Pamoate | 594.7 | −7.1 | −0.2 | −1.9 | −0.2 | −34.4 | antiparasitic |
| Cobicistat | 776.0 | −7.0 | −0.1 | −2.8 | −0.4 | −63.7 | P450 (e.g., CYP17) |

A considerable number of natural products, natural product derivatives, and biological molecules were present among the top hits. Plant flavonoids (quercetin, troxerutin, taxifolin), plant glycosides (sennosides A and B, digoxin), a citrus-derived sweetener (neohesperidin), and biological molecules (oxiglutatione, nadid) constituted about 36% of the top binding drugs with SARS-CoV-2 RdRP.

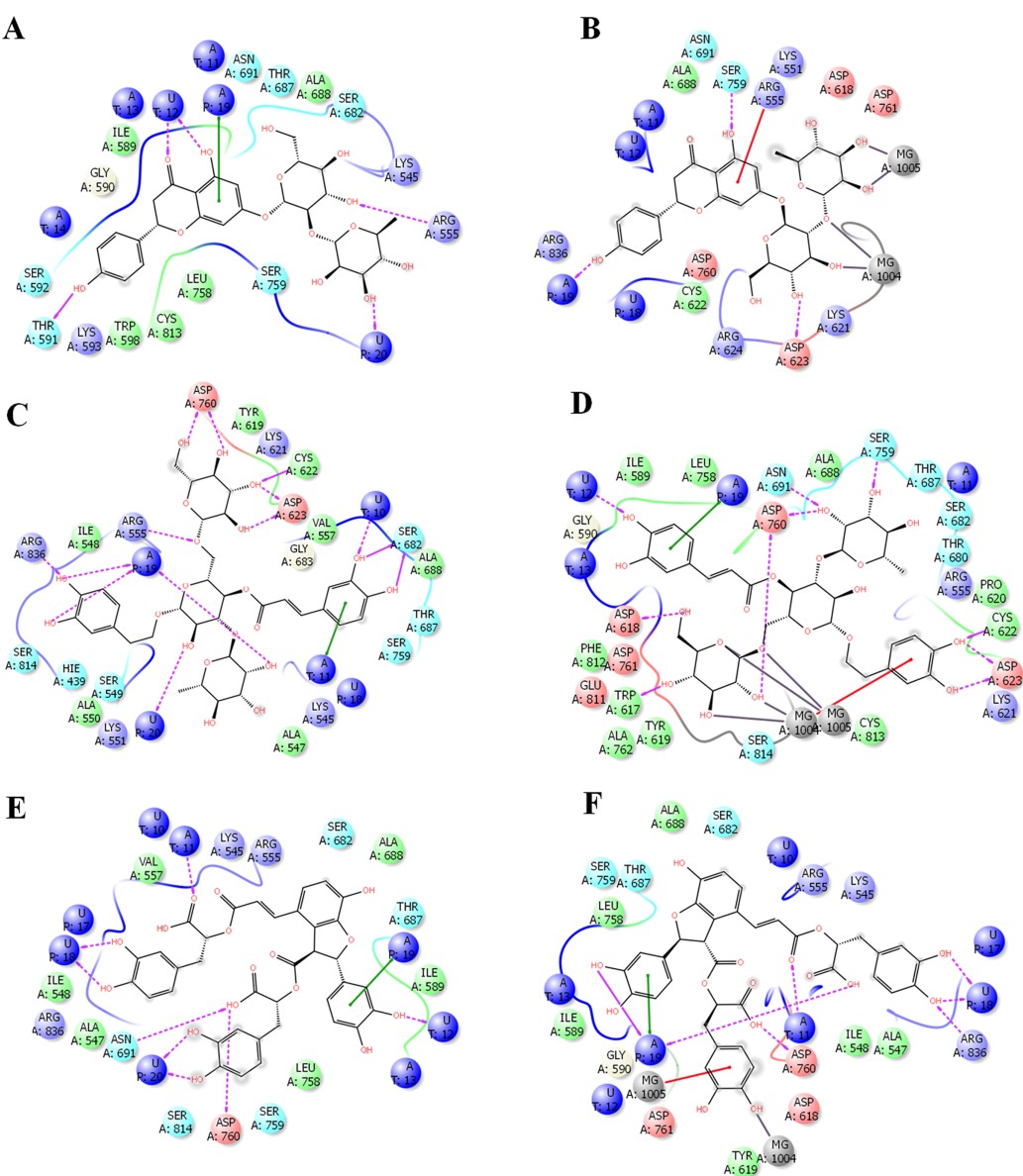

**Figure 2 Docking results in the presence or absence of magnesium ions.** The docking mode and ligand interactions of naringen, echinacoside and salvianolic acid with SARS-CoV-2 RdRP, in the presence or absence of magnesium ions. (A) Ligand interactions of naringen with RdRP in the absence of magnesium ions (B) Ligand interactions of naringen with RdRP in the presence of magnesium ions (C) Ligand interactions of echinacoside with RdRP in the absence of magnesium ions (D) Ligand interactions of echinacoside with RdRP in the presence of magnesium ions (E) Ligand interactions of salvianolic acid with RdRP in the absence of magnesium ions (F) Ligand interactions of salvianolic acid with RdRP in the presence of magnesium ions. Charged residue (negative) in pink, positive charged residue in blue, hydrophobic residues in cyan, hydrogen bonds: purple arrow, stacking interactions: green sticks, metal coordinates: gray sticks.

In order to get deeper insights into the interaction of drugs with SARS-CoV-2 RdRP, the binding mode and ligand interaction profiles of the drugs and also of remdesivir were examined (Fig. 2). The active site residues comprised positively charged amino acids (LYS476, ARG507, LYS681 and GLU694), negatively charged amino acids (ASP328,

**Table 2 Statistical correlation analysis of docking output.** Pearson's correlation of the obtained docking score with the drugs mw and interation parameters with SARS-CoV-2 RdRP.

| | Docking score vs. Mol. weight | Docking score vs. glide ligand efficiency | Docking score vs. glide lipo | Docking score vs. glide hbond | Docking score vs. glide evdw |
|---|---|---|---|---|---|
| Pearson $r$ | | | | | |
| $r$ | −0.08911 | 0.3242 | 0.1453 | 0.3442 | 0.1221 |
| 95% confidence interval | −0.17 to −0.04 | 0.28 to 0.37 | 0.098 to 0.19 | 0.3 to 0.38 | 0.07 to 0.17 |
| $R$ squared | 0.008 | 0.11 | 0.021 | 0.12 | 0.015 |
| $P$ value | | | | | |
| $P$ (two-tailed) | 0.0003 | <0.0001 | <0.0001 | <0.0001 | <0.0001 |
| Significant? (alpha = 0.05) | Yes | Yes | Yes | Yes | Yes |
| Number of XY Pairs | 1657 | 1657 | 1657 | 1657 | 1657 |

ASP501, ASP506, ASP643, ASP644 and ASP748), aromatic residues (TRP461, TYR474, TRP500, TYR502, TRP683 and PHE726), polar amino acids (GLN327, ASN330, SER475, SER565, SER642, CYS696, SER697, GLN698 and PRO715) and non-polar amino acids (GLY329, ALA313, ALA331, ILE333, VAL471, ILE472, GLY473, MET484, LEU641 and ALA645). Naringen formed two hydrogen bonds with ARG555 and THR591 (Fig. 2A). In contrast, echinacoside showed multiple interaction routes, including six hydrogen bonds with ARG555, CYC622, ASP623, SER682, ASP760 and ARG836 (Fig. 2B). It was noticeable that the number of hydrogen bonds formed between the docked ligands with the active is affected by the presence of magnesium ions. While metal coordinates is evident with the docked poses, this was underscored by lower hydrogen bonds interactions (Fig. 2). This might contributed to the observed lower docking scores in structures containing magnesium.

Statistical analysis revealed a significant positive correlation between the docking score and the hydrogen bond, lipophilic interaction, and ligand efficiency scores ($p < 0.001$; Table 2). There was a negative low but significant correlation between the obtained docking scores and the molecular weights of the compounds. This explains the superior docking scores of the selected set of drugs over the standard inhibitor remdesivir, which involved the lower molecular weights and improved hydrogen bonding and lipophilic interactions.

For further filtration to identify the most potent hits, compounds with a score of −7.00 or higher were subjected to XP-docking followed by MM/GBSA calculations (Table 3). Six compounds produced high docking scores exceeding −11.00: echinacoside, salvianolic acid B, ginsenoside, neohesperidin, troxerutin, and docetaxel. These top six were then used for further MD simulations.

## Molecular dynamics simulations for 20 ns

Combining docking with MD simulation is a powerful tool for drug discovery, as these techniques rank compounds on their binding affinity and can also evaluate target-receptor

**Table 3 The docking scores and estimated binding energy after XP-docking protocol.** The compounds were raked by their docking scores.

| Name | Docking score in the absence of Mg+2 | Docking scores in the presence of Mg+2 |
| --- | --- | --- |
| Echinacoside | −14.544 | −12.648 |
| Salvianolic acid B | −13.803 | −11.174 |
| Ginsenoside Re | −12.685 | −10.258 |
| Neohesperidin | −12.023 | −9.704 |
| Troxerutin | −11.465 | −9.174 |
| Docetaxel | −11.114 | −7.48 |
| Diosmin | −10.99 | −8.115 |
| Acarbose | −10.972 | −8.595 |
| Rutin (Rutoside) | −10.883 | −8.813 |
| Asiaticoside | −10.555 | −10.763 |
| Naringin (Naringoside) | −10.555 | −6.732 |
| Dihydrostreptomycin | −10.401 | −6.967 |
| Hesperidin | −10.389 | −9.286 |
| Neomycin sulfate | −10.387 | −8.159 |
| Fluvastatin sodium | −9.826 | −9.173 |
| Maltitol | −9.639 | −5.516 |
| Lactobionic acid | −9.383 | −6.431 |
| Amikacin disulfate | −9.327 | −7.947 |
| Isepamicin Sulphate | −9.244 | −7.946 |
| Oleuropein | −9.21 | −6.072 |
| Pravastatin sodium | −8.557 | −8.438 |

binding energy and drug-receptor dynamics. In this study, the top-ranked compounds from XP-docking were subjected to MD simulation followed by post dynamic analysis of binding energy. A two-step filtering process was adopted. First, the six compounds were run in MD simulation for 20 ns and their RMSD, RMSF, and MM/GBSA values evaluated. In the second stage, the top two compounds were tested in a more comprehensive 100 ns MD simulation. Structural changes in the backbone residues of RdRP relative to the initial structure (RMSD) were compared after 20 ns simulations (Fig. 3).

## MM/GBSA binding energies at 20 ns

According to the MM/GBSA binding energies, strong binding was indicated for docetaxel (−69.19 kcal/mol) and neohesperidin (−65.754 kcal/mol). Ginsenoside showed a moderate binding energy of −47.52 kcal/mol (Table 4). The other three compounds yielded low binding energy values, all less than −30.708 kcal/mol. In addition, all of the tested compounds showed low structure RMSD during the 20 ns simulations, with values <0.24 nm (Table 4). Based on the 20 ns simulation data, docetaxel and neohesperidin were selected for further analysis in 100 ns MD simulations.

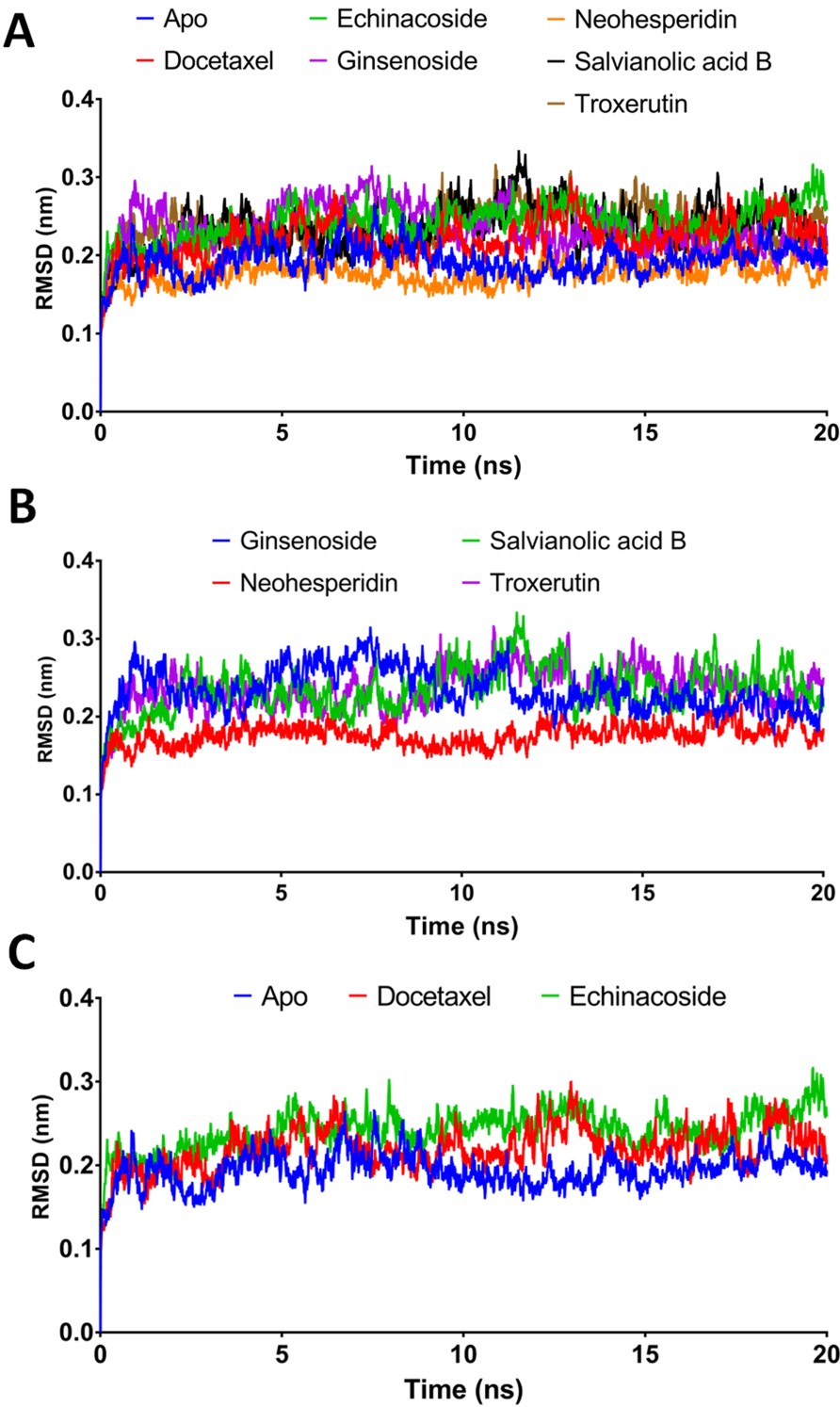

**Figure 3 Molecular dynamics simulation for 20 ns.** (A) The RMSD during 20 ns simulation of the top 6 compounds after XP-docking. (B) The RMSD of ginsenoside, salvianolic acid B, neohesperidin and troxerutin. (C) The RMSD of Apo RdRP, docetaxel and echinacoside. Apo implies SARS CoV-2 RdRP without any ligands.

**Table 4  Post dynamic analysis of RMSD.** Average RMSD and the MM-GBSA binding energy after 20 ns MD simulation for the top six compounds after XP-docking.

|  | Apo | Docetaxel | Echinacoside | Ginsenoside | Neohesperidin | Salvianolic acid B | Troxerutin |
|---|---|---|---|---|---|---|---|
| Average RMSD (nm) | 0.194 | 0.220 | 0.241 | 0.234 | 0.176 | 0.235 | 0.236 |
| MM-GBSA (kcal/mol) |  | −69.19 | −30.708 | −47.52 | −65.754 | −14.015 | −30.116 |

## Molecular dynamics simulations for 100 ns

In order to obtain deeper insight into the strongest-binding drugs, the two with binding energy >−60 kcal/mol (docetaxel and neohesperidin) were subjected to 100 ns MD simulations followed by analysis of binding free energy, RMSD, RMSF, hydrogen bond length, and radius of gyration. The estimated MM/GBSA binding energies were −67.273 and −63.669 for docetaxel and neohesperidin, respectively. These values imply superior binding for both of the compounds.

The 100 ns simulations yielded average RMSD values of 0.25 and 0.22 nm for docetaxel and neohesperidin, respectively. These values indicate stability of RdRP in complexes with docetaxel or neohesperidin. Both RMSD and per-residue RMSF showed almost similar profiles for RdRP complexes with docetaxel and neohesperidin (Fig. 4).

## Radius of gyration

Radius of gyration can be taken as a measure of the compactness of the system. Fig. 5 shows Rg variations observed during 100 ns MD simulations. For both compounds, the average Rg value was 3.18 nm. High Rg values indicate lower compactness or more unfolded protein, while low Rg indicates more stable structures. The similar Rg values indicate comparable stability of the two examined drugs when complexed with RdRP.

## Decomposition of MM/GBSA binding energy at 100 ns

Post-dynamic energy decomposition analysis was used to evaluate the dominant interactions during recognition of the drugs by RdRP (Table 5). The results revealed that van der Waals and electrostatic interactions were the predominant forces underlying binding of the drugs to RdRP. Specifically, van der Waals interactions were the major forces for both docetaxel and neohesperidin, followed by electrostatic interactions.

## The binding site of docetaxel and neohesperidin

The binding characteristics of docetaxel and neohesperidin with RdRP are provided in Fig. 6. The binding site is mostly composed of hydrophobic residues (ILE548, VAL557, ILE589, THR680, and ALA688), with the presence of few positively charged residues (LYS551, LYS545, and ARG555), negatively charged residues (ASP623 and ASP761), and neutral residues (SER549, SER682, ASN691, SER759, and SER814). The interaction of docetaxel and neohesperidin with RdRP was respectively supported by six and seven hydrogen bonds. Neohesperidin formed hydrogen bonding with ASP618, SER682, SER759, while docetaxel interacted with ILE548, SER549, LYS551 and SER814.

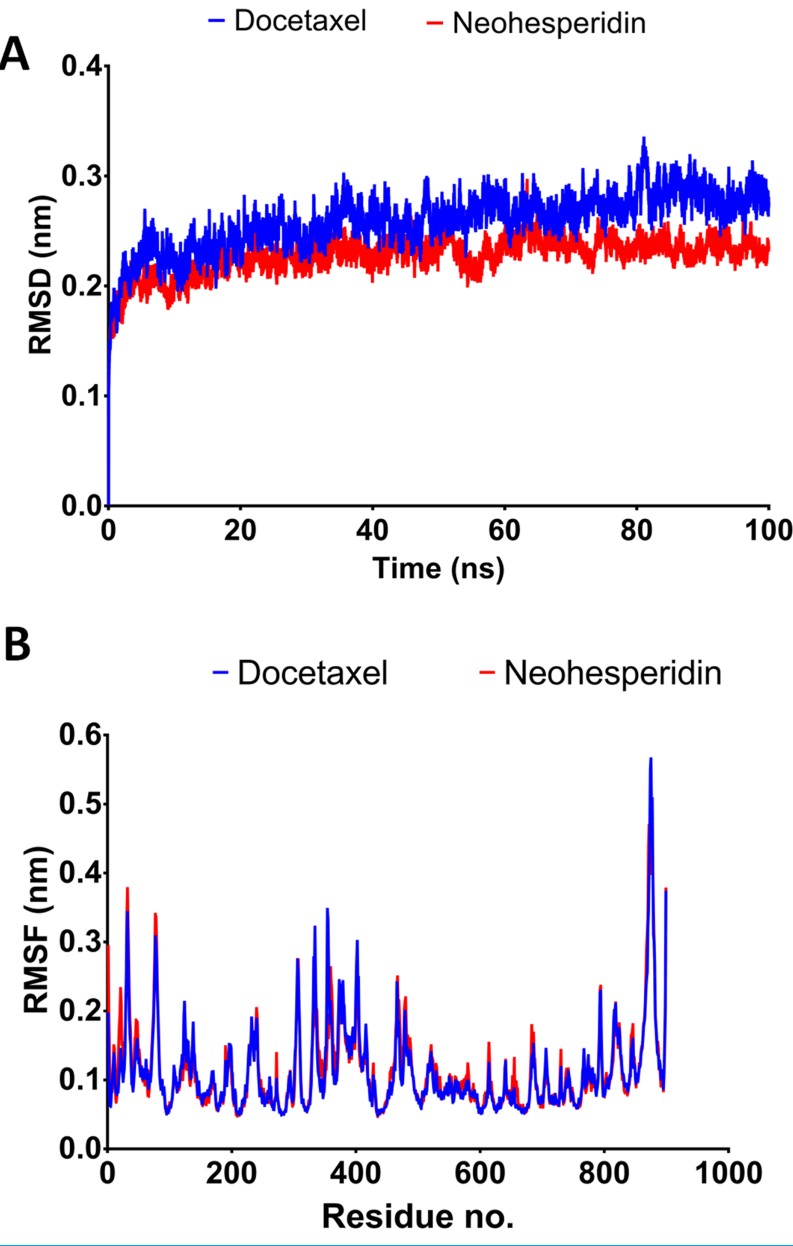

**Figure 4 Molecular dynamics simulation for 100 ns.** The RMSD during 100 ns simulation of the top two compounds after MD simulation for 100 ns, docetaxel and neohesperidin. (A) RMSD of docetaxel and neohesperidin. (B) The per-residue RMSF of RdRP bound with docetaxel and neohesperidin. The figure includes the average of three different experiments.

## The number of hydrogen bonds

The number of hydrogen bonds between docetaxel and neohesperidin and RdRP was traced during MD simulation (Fig. 7; Fig. S3). The average number of hydrogen bonds were 4.3 and 6.2 for docetaxel and neohesperidin, respectively. Additionally, the distance between the two drugs and two selected residues in the active site was traced during simulation (Fig. 7B; Fig. S4). The selected residues were LYS521 for docetaxel and ASP588

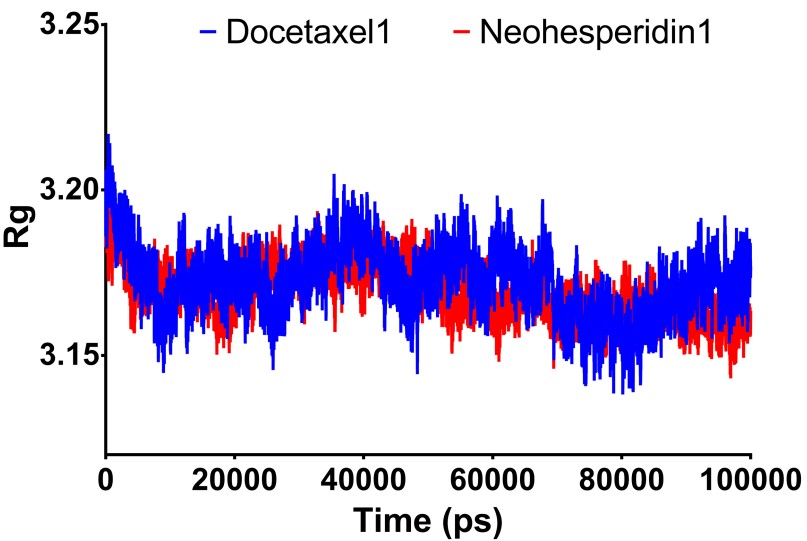

**Figure 5 Molecular dynamics simulation for 100 ns.** Radius of gyration of RdRP bound with docetaxel and neohesperidin after 100 ns molecular dynamics simulation. The figure includes the average of three different experiments.

**Table 5 Energy decomposition analysis.** Decomposition of the estimated MMGBSA binding energy for the binding of docetaxel and neohesperidin with SARS CoV-2 RdRP.

|  | Docetaxel | Neohespridin |
|---|---|---|
| Van der Waal energy | −158.953 | −169.743 |
| Electrostattic energy | −19.672 | −111.149 |
| Polar solvation energy | 130.971 | 238.858 |
| SASA energy | 19.581 | −21.655 |
| Binding energy | −67.273 | −63.669 |

for neohesperidin. After initial stabilization within the first few ns, the results indicated the lack of large fluctuations in the position of both drugs in the active site.

## DISCUSSION

This work covers a knowledge gap about the SARS-CoV-2 RdRP and its relation to the two previous fatal CoVs, SARS-CoV and MERS-CoV. Additionally, the structure of RdRP was targeted by a set of FDA-approved drugs for the purpose of relocating already approved and safe drugs for SARS-CoV-2 treatment.

Based on their pharmacokinetic aspects, some of these top hits are not appropriate for systemic application due to their low levels of absorption for example, sennosides (*Leng-Peschlow, 1986*) and pyrantel pamoate (*Ryan, 2018*). Others have specific applications with an impact on specific body systems or disorders, such as digoxin (cardiovascular system), acarbose (antidiabetic), venetoclax and taxifolin (anticancer drugs), and chlorhexidine (a drug for topical application).

Ritonavir is the only antiviral predicted to bind with SARS-CoV-2 RdRP. It is approved for HIV treatment and acts by inhibition of the virus protease (*Croxtall & Perry,*

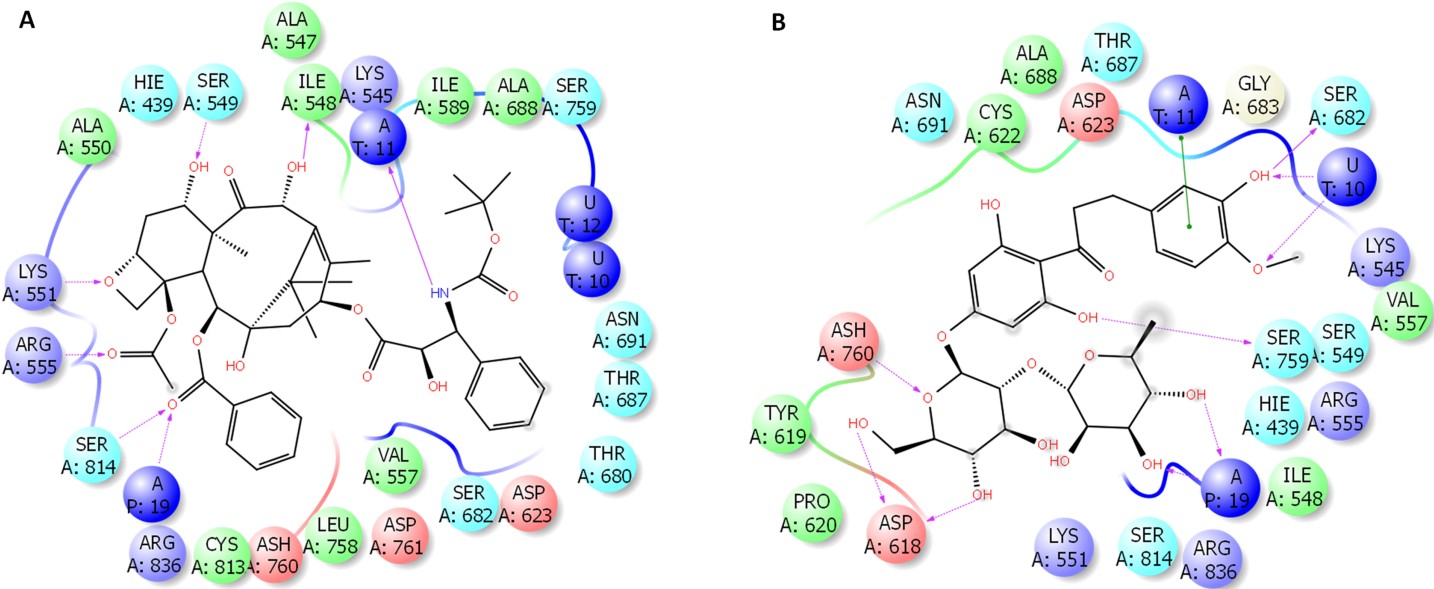

**Figure 6 Molecular dynamics simulation for 100 ns.** The ligand interactions of docetaxel and neohesperidin with SARS CoV-2 RdRP from the average structure after 100 ns molecular dynamics simulation. (A) Docetaxel (B) Neohesperidin.    

_2010_). Ritonavir was in the 6th rank of the top hits. A recent clinical report showed the improvement of an index case with pneumonia with a significant decrease in the virus load after ritonavir/lopinavir treatment (_Lim et al., 2020_). The drugs with higher docking scores, such as nadid or thymopentin, are general body and immune stimulants that might be of value in COVID-19 treatment.

Remdesivir was used as a reference drug for binding with RdRP. It successfully inhibited MERS-CoV RdRP. The docking output of remdesivir had a score of −5.98, which was lower than the scores of the 26 drugs selected that had a high affinity for RdRP.

Docetaxel complex with SARS-CoV-2 RdRP showed high stability and free binding energy. This interaction is driven by favorable van der Waals and electrostatic interactions, and includes the formation of about six hydrogen bonds with RdRP active site residues. By MD simulation for 100 ns, docetaxel formed an average of 4.3 hydrogen bonds. Docetaxel has large mw (807.9 g/mol). However, this drug obeys the Lipiniski's rule of five in LogP and favorable hydrogen donor count (5 or lower). In addition, docetaxel is a cytotoxic agent with clinical approval in the treatment of different types of cancers, for example, breast and prostate cancers (_Lyseng-Williamson & Fenton, 2005_; _Tannock et al., 2004_).

Neohesperidin is a flavanone glycoside from citrus fruits with wide applicability in vascular diseases and cancer treatment trials (_Walker, Janda & Mollace, 2014_). Similar to docetaxel, a favorable hydrogen bonding profile and electrostatic and van der Waals interactions support the potent binding of neohesperidin with SARS CoV-2 RdRP. By MD simulation for 100 ns, neohesperidin formed an average of 4.3 hydrogen bonds.

Taking the docking and MD simulation results together, the two compounds exhibited favorable binding features comprising low RMSD, high binding free energy, low radius

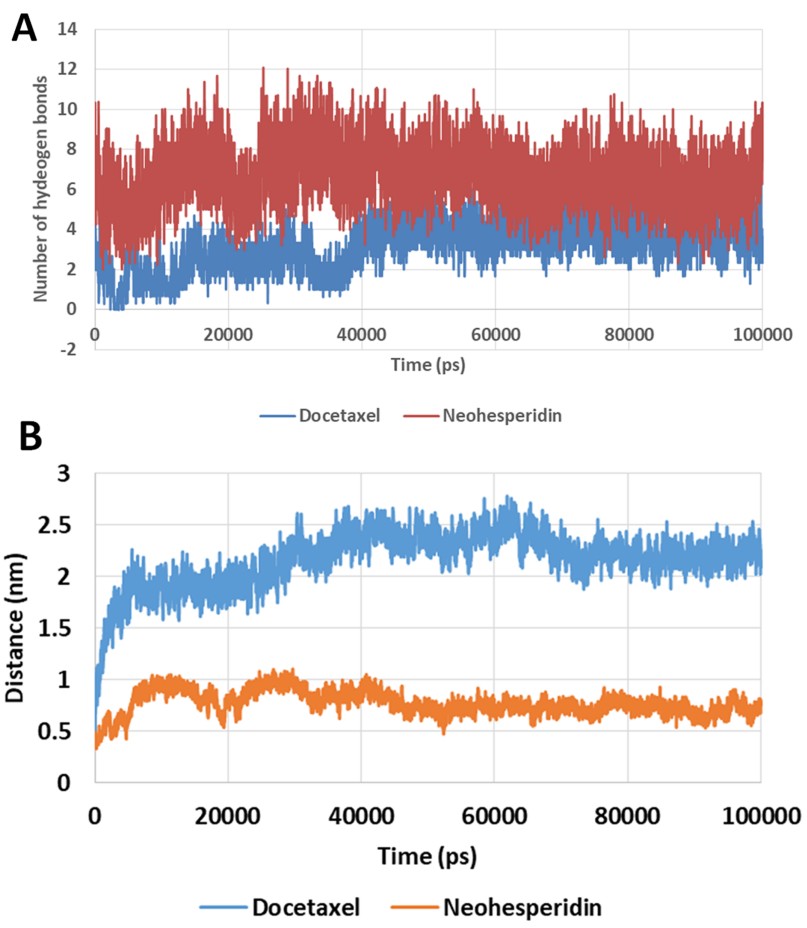

**Figure 7 Molecular dynamics simulation for 100 ns.** The number of hydrogen bonds and distance from RdRP residues during 100 ns simulation of the top two compounds after MD simulation for 100 ns, docetaxel and neohesperidin. (A) The number of hydrogen bonds formed by docetaxel and neohesperidin (B) The distance between the two drugs and two selected residues in the active site during 100 ns MD simulation. The selected residues were LYS521 for docetaxel and ASP588 for neohesperidin.

of gyration, formation of several hydrogen bonds, strong electrostatic interactions, and the presence of polar and hydrophobic components in the interaction with RdRP. Based on these findings, docetaxel and neohesperidin are most likely to be strong inhibitors of SARS CoV-2 RdRP.

## CONCLUSIONS

After the molecular modeling and virtual screening of approximately 1,664 FDA-approved drugs, a set that interact strongly with SARS CoV-2 RdRP has been formed and is provided. These are suggested for use as an addition to the standard treatment.

The superiority of obtained values over a known MERS-CoV inhibitor further supports the merit of repurposing these compounds against SARS CoV-2 RdRP. Furthermore, the binding potency and molecular dynamics of docetaxel and neohesperidin in particular recommends them for such use. However, further assessment by antiviral assays is required.

## ABBREVIATIONS

| | |
|---|---|
| **FDA** | Food and Drug Administration |
| **SARS-CoV-2** | Severe Acute Respiratory Syndrome Coronavirus-2 |
| **COVID-19** | coronavirus disease 2019 |
| **RdRP** | RNA-dependent RNA polymerase |
| **MERS CoV** | Middle East Respiratory Syndrome Coronavirus |
| **S** | spike |
| **E** | envelope |
| **M** | membrane |
| **N** | nucleocapsid |
| **GISAID** | Global initiative on sharing all influenza data |

## ACKNOWLEDGEMENTS

The authors extend their appreciation to the Deputyship for Research & Innovation, Ministry of Education in Saudi Arabia for funding this research work through the project number IFT20110.

### Funding

This project is funded by the Deputyship for Research & Innovation, Ministry of Education in Saudi Arabia for funding this research work through the project number IFT20110. The funders had no role in study design, data collection and analysis, decision to publish, or preparation of the manuscript.

### Grant Disclosures

The following grant information was disclosed by the authors:
Deputyship for Research & Innovation, Ministry of Education in Saudi Arabia: IFT20110.

### Competing Interests

The authors declare that they have no competing interests.

### Author Contributions

- Mahmoud Kandeel conceived and designed the experiments, performed the experiments, analyzed the data, prepared figures and/or tables, authored or reviewed drafts of the paper, and approved the final draft.
- Yukio Kitade conceived and designed the experiments, performed the experiments, analyzed the data, prepared figures and/or tables, authored or reviewed drafts of the paper, and approved the final draft.
- Abdullah Almubarak conceived and designed the experiments, performed the experiments, analyzed the data, prepared figures and/or tables, authored or reviewed drafts of the paper, and approved the final draft.

## Data Availability

Raw data is available in the Supplemental Files.

## Supplemental Information

Supplemental information for this article can be found online at http://dx.doi.org/10.7717/peerj.10480#supplemental-information.

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
