# Peer review of "Repurposing FDA-approved phytomedicines, natural products, antivirals and cell protectives against SARS-CoV-2 (COVID-19) RNA-dependent RNA polymerase"

_PeerJ, doi:10.7717/peerj.10480_

## Round 0.1 · original submission · Major Revisions

Dear Dr. Kandeel and colleagues:

Thanks for submitting your manuscript to PeerJ. I have now received two independent reviews of your work, and as you will see, the reviewers raised some concerns about the research. Despite this, these reviewers are optimistic about your work and the potential impact it will have on research studying the repurposing of FDA-approved phytomedicines, natural products, antivirals and cell protectives against SARS-CoV-2 (COVID-19) RNA-dependent RNA polymerase. Thus, I encourage you to revise your manuscript, accordingly, taking into account all of the concerns raised by both reviewers.

Please also ensure that your figures and tables contain all of the information that is necessary to support your findings and observations. It appears Figure 3 may not be useful as presented. The Methods should be clear, concise and repeatable. Please ensure this. The docking experiments and alignments should be straightforward. Please address all of the typos,

While the concerns of the reviewers are relatively minor, this is a major revision to ensure that the original reviewers have a chance to evaluate your responses to their concerns. There are many suggestions, which I am sure will greatly improve your manuscript once addressed.

I look forward to seeing your revision, and thanks again for submitting your work to PeerJ.

-joe

Reviewer 1 ·

Basic reporting

English language and style are fine/minor spell check required

Abstract & Introduction:

Line 26 & 57 - coronavirus disease 19 (COVID-19) should instead be coronavirus disease 2019 (COVID-19)

Experimental design

- Protein sequencing data is repetition as its already documented at >96% similarity in literature
- In Figure 4 and 8, There are few interactions that are not commented on
- Please provide detail of the docked complex for Figure 4 (i.e, which atom or functional group is interacting with the residue)
- Justify while molecular weight of compound > 600 g/mol can be considered as a drug agent for the proposed infection
- In line 368 and the capture under Figure 4 change hrdrophobic residues to hydrophobic residues and stocking to stacking interaction

Validity of the findings

The article is interesting in light of CADD assistance in predicting probable solutions to the current pandemic which is a hot topic right now. Although not all of the top hits were appropriate due to pharmacokinetic aspects the information does however provide possible solutions in the form of docetaxel and neohesperidin.

Reviewer 2 ·

Basic reporting

No comment

Experimental design

There are some points that the authors should make clearer.

The RdRP crystal has two Mg2+ ions in the vicinity of the inhibitor that are important for the catalytic activity. Were these ions considered during docking and dynamics calculations or not? If not, why?

The alignment of the sequences is not applied in the work; apart from saying that the RdRp sequence of Sars-CoV-2 is extremely similar to that of Sars-CoV and slightly different from Mers, it brings no added value. Although very limited, the differences between the two RdRp sequences Sars-Cov and Sars-CoV-2 are found above all near the active site? If so, these small differences could be significant.

Is the Remdesivir structure that has been considered the crystallographic one or does it represent a docking pose? In the first case, the authors should explain how they know if the docking procedure they have chosen is reliable or not; in the second case, what is the Remdesivir docking score with SP and XP?

Figure 3 of docking results is of little use. Too many superimposed molecules do not allow us to understand if the respective poses are right or not and it is not possible to understand the relative position of the molecules with respect to Remdesivir, which was then taken as a reference. Also, the SP docking results are completely different from the XP ones. The molecules in the two rankings are different as well as the scores. We agree that the two scoring functions are different, but in this case it seems that two completely different scenarios are considered ...).

How many replicas of the various dynamics have been performed? (Maybe one is insufficient ...).

Table 5 shows the decomposition of the estimated MMGBSA binding energy of one of the two molecules selected for extended dynamics. It would be interesting if the authors also showed the other molecule, in order to have a complete picture of the final result of the 100ns dynamics and of the behavior of the two molecules in complex with RdRp.

Figure 8 shows the interactions of the two molecules within the active site. Are these interactions those present at the last frame of the dynamics or do they represent some other frame? Are the interactions held constant throughout the simulation or are they formed during molecular dynamics? It would be interesting, especially for docetaxel, to see how this compound positions itself inside the binding pocket.

Validity of the findings

No comment

Additional comments

This reviewer would like to point out also some minor changes:

Figure 4: check if the amino acid residues involved in hydrogen bond interactions and highlighted in this Figure are indeed the same reported in the text (line 164);

correct hrdrophobic and stocking in the legend of Figure 4;

line 216 and throughout the text I would suggest reporting the acronym vdw as VDW, or even better in full as van der Walls interaction(s).

---

## Round 0.2 · accepted · Accept

Dear Dr. Kandeel and colleagues:

Thanks for revising your manuscript based on the concerns raised by the reviewers. I now believe that your manuscript is suitable for publication. Congratulations! I look forward to seeing this work in print, and I anticipate it being an important resource for groups studying the repurposing of FDA-approved phytomedicines, natural products, antivirals and cell protectives against SARS-CoV-2 (COVID-19) RNA-dependent RNA polymerase. Thanks again for choosing PeerJ to publish such important work.

Best,

-joe